# Effectiveness of Mindfulness-Based Interventions with Respect to Psychological and Biomedical Outcomes in Young People with Type 1 Diabetes: A Systematic Review

**DOI:** 10.3390/healthcare12181876

**Published:** 2024-09-19

**Authors:** Tamara Gutiérrez-Domingo, Naima Z. Farhane-Medina, Joaquín Villaécija, Sebastián Vivas, Carmen Tabernero, Rosario Castillo-Mayén, Bárbara Luque

**Affiliations:** 1Maimonides Biomedical Research Institute of Cordoba (IMIBIC), 14071 Córdoba, Spain; nfarhane@uco.es (N.Z.F.-M.); jvillaecija2@uco.es (J.V.); svivas@uco.es (S.V.); carmen.tabernero@usal.es (C.T.); rcmayen@uco.es (R.C.-M.); bluque@uco.es (B.L.); 2Department of Psychology, University of Cordoba, 14071 Córdoba, Spain; 3Reina Sofia University Hospital, 14071 Córdoba, Spain; 4Department of Psychology, University of Salamanca, 37007 Salamanca, Spain; 5Institute of Neurosciences of Castilla y León (INCYL), University of Salamanca, 37007 Salamanca, Spain

**Keywords:** diabetes mellitus type 1, youth adult, young, mindfulness, psychological well-being, glycaemic control

## Abstract

Background: Type 1 diabetes is a chronic disease especially affecting young people. Mindfulness-based psychological interventions might reduce emotional symptoms post-diagnosis, but the evidence is limited. Objectives: This systematic review aimed to evaluate the effectiveness of mindfulness interventions on psychological well-being and biomedical variables in young people with type 1 diabetes. Methods: A systematic review of trials was conducted that involved a bibliographic search in electronic databases (Web of Science, MEDLINE, SciELO, Scopus, PsycINFO, and Cochrane Library) considering studies published between 2013 and 2024. Results: A total of 434 records were identified, of which 252 underwent selection according to title and abstract, leaving 32 that were evaluated for eligibility and 7 included in this review. From Google Scholar, six more studies were identified and evaluated, and two were selected. Finally, nine studies were subjected to full reading and a detailed analysis of the inclusion criteria. A total of 66.6% of the studies were evaluated as having a methodological quality of moderate or optimal, but the samples analysed tended to be small, and only two articles carried out short-term follow-up evaluations. Conclusions: Mindfulness-based interventions, upon reviewing the preliminary results, may be posited as a viable strategy to enhance psychological (anxiety, diabetes distress, perceived stress, depression, self-efficacy, psychological well-being, and quality of life) and biomedical outcomes (glycaemic control, blood glucose levels, and diastolic blood pressure) for type 1 diabetes in young people. Although promising, further research is required to improve the quality, methodology, and design of studies.

## 1. Introduction

Diabetes is a chronic disease that requires regular medical care and self-management education to prevent acute complications and reduce the risk of long-term complications [1]. Diabetes care is a multifaceted process that extends beyond glycaemic control. Indeed, successful diabetes care demands a systematic approach to support the behaviour of patients, focusing on change efforts and incorporating high-quality diabetes self-management education and support [2]. The new Global Pact against Diabetes of the World Health Organization [3] is particularly pertinent, as it strives to advance efforts aimed at ensuring treatment accessibility for the entire population at risk. Within this framework, it seems necessary to deepen research on the reduction of both biomedical and psychological adverse outcomes for patients with diabetes.

Type 1 diabetes (T1D) becomes a chronic disease that requires a daily administration of insulin and continuous self-monitoring [4]. Even though T1D can occur at any age, it is one of the most frequent chronic diseases in childhood, adolescence, and young adulthood, and it has an increasing incidence rate [5]. The health of young people aged 10–24 years [6] has been largely neglected [7]. Globally, the estimated prevalence of T1D in the age group of 0–19 years is 1,211,900 [5]. The annual number of new cases (i.e., incidence) is approximately 1.52 million in people under 20 years old [8]. In the future, according to information from the Global Burden of Diseases study [9], the age-standardized global prevalence of T1D is expected to increase by 23.9%, from 0.2% in 2021 to 0.3% in 2050. The heightened prevalence of T1D among individuals diagnosed at an age younger than 30 years raises concerns, as this demographic faces a mortality risk up to five times higher than their counterparts without diabetes [10]. This underscores the critical need for effective management strategies, comprehensive care, and targeted interventions to improve outcomes and reduce the impact of T1D on the overall health and psychological well-being of young people.

The primary biomedical result, frequently analysed in patients with diabetes through the study of blood glucose levels, is metabolic and glycaemic control. Specifically, the parameter widely analysed in studies of patients with T1D, type 2 diabetes (T2D), and prediabetes, at a biomedical level, is glycosylated haemoglobin (HbA1c). This consists of a blood test that measures the average level of glucose in the blood during the last three months, which contributes to the diagnosis and establishes the proper management of diabetes. Another relevant biomedical factor related to T1D is blood pressure (BP). In fact, the overriding causes of premature mortality in patients with T1D are vascular complications, which are aggravated by comorbid diseases like hypertension [11]. BP is understood as the tension exerted by circulating blood on the walls of blood vessels, and it involves two measurements: systolic BP and diastolic BP [12]. In addition to the physical symptoms associated with T1D, living with a chronic long-term disease can have significant emotional impacts, especially during adolescence [13]. Adolescence is a critical period marked by physical, emotional, and social changes, and the presence of a chronic condition introduces additional complexities. Psychosocial factors play a pivotal role during this stage, influencing how adolescents and young people cope with and manage their diabetes [14]. Psychological factors could be behind the metabolic control of the disease, and both should be understood as a process of constant multidirectional exchange that affects normal development during childhood and adolescence [15].

In particular, psychological symptoms linked to depression and anxiety have been estimated to be elevated in adolescents with T1D [16,17]. Likewise, stress specifically related to T1D in adolescence is also often recurrent [18]. Diabetes distress (DD) is understood as reports of sensations of social distress and fears related to diabetes, linked to specific tasks related to the control of the disease by the patient themselves and their main caregivers, and DD affects well-being and clinically relevant measures of glycaemic control [19]. To counteract the psychological symptoms associated with T1D in adolescence, the promotion of emotional well-being and its self-regulation seems to play a key role. Emotional well-being is understood as the emotional quality associated with individual daily experiences [20], such as the frequency and intensity of experiences of joy, fascination, anxiety, sadness, anger, and affection that cause life to be interpreted as nice or nasty. Similarly, the proper management of diabetes requires correct self-regulation behaviour on the part of adolescents [21]. For its part, health-related quality of life (HRQoL) is a relevant indicator of general health status when studied in the adolescent population with T1D [22]. Adolescents with T1D have lower HRQoL scores, which affects metabolic control [23]. In turn, poorer metabolic control in adolescents with T1D has been associated with lower HRQoL [24].

Given this panorama, psychoeducational interventions for adolescents with T1D have been proposed to improve knowledge and coping with the disease [25]. Mindfulness-based interventions (MBIs) are gaining special attention, and with increasing effects, as indicated by a recent systematic review of an adult T2D population in which MBIs’ effectiveness was shown [26]. Previously, Noordali et al. [27] pointed out that MBIs have shown to be effective in reducing both biomedical (glycaemic control and BP) and psychological (symptoms of depression, anxiety, and distress) complications in adults with T1D and T2D, at least in the short term. Subsequently, a systematic review and meta-analysis conducted by Ni et al. [28] found that MBIs appear to have benefits for T1D in depression, stress, and diabetes-related anxiety (i.e., DD). Closely reviewing the scientific publications on this topic should shed more light on this field of study.

MBIs are based on the practice originally proposed by Kabat-Zinn et al. [29] for patients with chronic pain. MBIs are a group of effective interventions based on the practice of mindfulness, which is focused on promoting the deliberate and non-judgmental awareness of the present experience, including specific components such as mindful breathing techniques, body scanning, and meditation, with the aim of regulating stress and reducing negative effects in a wide range of physical and mental health problems [30]. The most widely used by T1D programmes are Mindfulness-Based Stress Reduction (MBSR), Mindfulness-Based Cognitive Therapy (MBCT), Mindfulness-Based Eating Awareness Training (MB-EAT), and Acceptance and Commitment Therapy (ACT) with mindfulness incorporated in some of its principles and techniques [31,32,33]. Recently, Inverso et al. [34] conducted a review of the literature on the possible psychosocial and health benefits of MBIs for young people with T1D and T2D, finding a reduction in the symptoms of depression and anxiety due to diabetes, although the results were inconsistent regarding the physical health benefits.

In short, in light of the results of recent integrative reviews [34,35], the effects of MBIs are promising. They have been shown to be effective for adult patients with diabetes. However, the scientific literature regarding young people is still scarce. Therefore, future research with young people with diabetes might include properly adapted MBIs. In this line, promoting emotional well-being and self-regulation through psychological interventions based on practices of mindfulness could influence the T1D self-management behaviours of young people, and the effects might boost the perception of HRQoL. Our main objective was thus to explore the impact of MBIs as a regulation strategy for biomedical outcomes (glycaemic control and BP) and psychological variables (emotional well-being, anxiety and depression, self-efficacy, DD, and HRQoL) in young patients with T1D. 

## 2. Materials and Methods

### 2.1. Study Design

We conducted a systematic review in accordance with the Preferred Reporting Items for Systematic Reviews and Meta-Analysis (PRISMA; Appendix A) [36] guidelines. Additionally, the systematic review protocol was officially registered on the International Prospective Register of Systematic Reviews (PROSPERO) platform (ref. CRD42023467044) [37].

### 2.2. Search Strategy

The aim of this study was to investigate the effectiveness of MBIs for young populations with T1D, evaluating their impact on biomedical parameters and relevant psychological variables. Prior to initiating the database search strategy, we agreed on specific technical inclusion criteria, which resulted in the formulation of the PICOS question [38] (see Table 1). Then, our systematic search was carried out by four authors working independently (T.G.-D., N.Z.F.-M., J.V., and S.V.) in six consolidated international databases (Web of Science, MEDLINE, SciELO, Scopus, PsycINFO, and Cochrane Library) in order to identify relevant studies published during the last 11 years in the period between January of 2013 and July of 2024. Google Scholar was also consulted as an additional source. The search terms used for this review were (Adolescen* OR Teen* OR Youth* OR “Young people”) AND (diabetes OR T1D* OR idd*) AND (Mindful* OR Mind-Body* OR MBI OR MBSR OR MBCT OR MB-EAT OR Meditation*). To refine the search strategy, several search strings were used in the selected databases, resulting in a large number of results (Appendix A).

### 2.3. Inclusion and Exclusion Criteria

This review’s inclusion criteria can be seen in Table 2.

### 2.4. Selection Process

Following the screening and application of our selection criteria for data collection, the initial step involved eliminating duplicate articles using Zotero software version 6.0.20. Subsequently, the selected studies underwent a comprehensive review and categorization by four researchers working independently (T.G.-D., N.Z.F.-M., J.V., and S.V.) based on title and abstract, utilizing a predefined template designed to compile information from the included studies. Disagreements were solved by discussion and consensus.

### 2.5. Data Extraction

Finally, the selected full-text articles were peer-reviewed, and a comprehensive set of data was extracted into an Excel version 2408 spreadsheet. The list included various details like author(s), year of publication, country in which the study was conducted, information about the study design (including whether the studies were pilot studies or included follow-up evaluations and their respective duration), sample characteristics, variables analysed, intervention characteristics, and main outcomes. The data were extracted by four independent authors (T.G.-D., N.Z.F.-M., J.V, and S.V). Again, discrepancies in the extracted data were addressed through discussion and consensus among all investigators at meetings.

### 2.6. Methodological Quality Assessment of the Studies Included (Risk of Bias)

The methodological quality of the selected studies was analysed using the National Institutes of Health (NIH) Quality Assessment Tool [39]. Two versions of this tool were required, one for controlled intervention studies and another for before–after (pre–post) studies with no control group. Each tool included 14 or 12 items, respectively, each of which can be marked as yes (score 1), no (score 0), or not reported/cannot determine (score 0). To evaluate the final scores qualitatively, studies with scores above 11 were classified as ‘good’, those below 6 were considered ‘poor’, and studies within the range of 7 to 10 were categorized as ‘fair’. The evaluations were conducted independently by two researchers (T.G.-D. and N.Z.F.-M.). In cases of discrepancies, a third evaluator (S.V.) reviewed the study.

## 3. Results

### 3.1. Study Selection

Figure 1 shows the study selection process used in this systematic review. Initially, 434 records were found through the primary search. After removing duplicates, 252 records were screened based on their titles and abstracts. Subsequently, 220 records were excluded based on our selection criteria, leaving 32 records for a more comprehensive analysis. Finally, after a full-text examination, seven studies met the inclusion criteria for this review. Furthermore, a supplementary search was conducted in other databases with less systematic possibilities, such as Google Scholar. This led to the inclusion of six studies, of which two met the review’s screening criteria. In total, nine studies were included [40,41,42,43,44,45,46,47,48].

### 3.2. Quality Assessment

Of the eight studies assessed using the Controlled Intervention Studies Tool [39], the majority (44.4%, *n* = 4) were categorized as having ‘fair’ methodological quality [40,43,45,46]. Three studies were evaluated as ‘poor’ (33.3%) [41,44,47], while only one study satisfied most of the criteria and received a ‘good’ rating [42]. The remaining study was evaluated using the Before–After (Pre–Post) Studies With No Control Group scale and received a ‘fair’ rating for its methodological quality [48]. In summary, 66.6% of the studies included (*n* = 6) were considered to have a high or acceptable level of methodological quality [40,42,43,45,46,48], while the rest appeared to have a potentially high risk of bias [41,44,47]. Additional information on the evaluation of each study can be found in Table 3.

### 3.3. Characteristics of the Studies

Table 4 summarises the key characteristics of the studies. Most of the studies included in this systematic review were conducted in Asia (*n* = 4), followed by North America (*n* = 3), Europe (*n* = 1), and Oceania (*n* = 1). The studies were mostly conducted within the last 5 years (*n* = 7). In terms of study design, eight of the nine studies included in the analysis employed an interventional design with a control group [40,41,42,43,44,45,46,47], while one used a before–after (pre–post) design without a control group [48]. Regarding the sample, the mean sample size was 35 participants, ranging from 6 to 64 (SD = 19.29). The age of the participants ranged from 9.72 to 24.8 years (M = 15.42, SD = 4.23), and the sex distribution was almost equal, with 44.1% of the sample being female. Most of the studies (66.6%) were pilot studies, and only two studies included a follow-up measurement [42,47]. 

The studies described interventions that lasted an average of 9.89 weeks, ranging from 3 to 24 weeks. Each session lasted approximately 1.39 h, with a total intervention duration of around 10.44 h. Regarding adherence to the intervention, the studies reported a retention rate of 77.1% (SD = 9.86). Per the analysed variables, the details of the instruments used are presented in Table 5. Six studies included both psychological and biomedical variables [40,41,42,46,47,48], while the remaining studies exclusively focused on psychological variables. Regarding the MBI protocol of the interventions, only a few descriptive characteristics could be extracted due to the heterogeneity of the interventions. These characteristics are detailed in Table 4. All the interventions were group-based, except for one that took place online [47], although this could also have included interaction tasks for the participants. Designs based on MBSR interventions (*n* = 3) [42,47,48] and ACT (*n* = 3) [40,44,45] were predominant.

### 3.4. Effectiveness of the Interventions

The studies yielded several encouraging outcomes with respect to the psychological factors under investigation. The studies that evaluated psychological variables found both between-group [40,41,44,45,47] and within-group differences [42,46,48] after the interventions (see Table 4 for details). MBIs were shown to effectively reduce psychological distress, stress, depression, and anxiety symptoms, as well as alleviate feelings of guilt and diabetes-related distress [40,42,44,45,46,48]. However, one of the studies found no improvement in depression [40,42]. One study, comparing three types of interventions, found that an MBI improved stress levels, while support groups improved glycaemic control and depressive symptomatology [42].

Furthermore, these interventions were shown to enhance psychological well-being by increasing self-efficacy, promoting greater flexibility and adaptability in dealing with the disease, and improving overall life satisfaction [40,44,45], although some studies did not find improvements in quality of life or in any of its sub-factors [40,46]. While the number of studies that investigated the effects of MBIs on biomedical factors was somewhat limited, the available research showed promising results both between groups [40,47] and within groups [46,48] regarding certain relevant variables. These findings reflected significant improvements in HbA1c [40,47], blood glucose levels [46,48], and diastolic BP [46]. However, no significant improvements were found in body mass or systolic BP [46]. Moreover, significance was not always obtained per the HbA1c variable [42], perhaps due to the lack of long-term follow-up. 

Finally, in the studies where data on acceptability and feasibility were reported, high levels of retention were found in most [40,41,42,45,48], except in two studies [42,45]. Aditionally, high levels of satisfaction were found in all three studies that evaluated it [40,41,48].

## 4. Discussion

This study’s review question pertained to the possible impact of MBIs on psychological well-being and biomedical variables (especially HbA1c) in young people with T1D. As proposed by Luque et al. [25], psychoeducational interventions in young people with T1D might improve knowledge and coping with the disease, and we consider addressing the emotional impact of T1D as crucial for comprehensive care. In this sense, support from healthcare professionals, psychologists, and a strong support system can be beneficial in helping individuals cope with the emotional aspects of living with diabetes. Our review of the scientific literature showed that MBIs have been used for a wide range of physical and mental health problems, with promising results [30]. Specifically, MBIs have been explored in adult populations with T2D, where their efficacy was demonstrated in terms of psychological and biomedical results, objectively reducing distress and HbA1c levels and promoting self-care [26].

The explored results can also be discussed regarding the psychological and biomedical outcomes and type of specific MBI program. The studies analysed showed some psychological results with significant changes after interventions based on mindfulness programs for young people with T1D. In 2021, Ni et al. [28] found that MBIs appeared to have benefits for diabetes-related depression, stress, and anxiety, and the authors recommended further literature reviews to shed more light on these findings. Specifically, after reviewing our results, we noted a reduction in anxiety [40], DD [48], perceived stress [42,44], self-efficacy [44], and depression [45] and increased psychological well-being [45] and quality of life [47] in young people with T1D. Also, the studies analysed in this review reported some biomedical outcomes with significant changes after interventions based on programs using mindfulness, especially regarding HbA1c levels [40,46] in addition to mean blood glucose and diastolic BP improvements [46]. These results are in line with Noordali et al. [27], who claimed that MBIs had not only been shown to be effective in reducing some psychological symptoms (symptoms of depression, anxiety and distress) but also in improving biomedical (glycaemic control and BP) variables in adults with T1D and T2D.

Finally, regarding the specific type of MBI that was used in the different studies, the findings seemed to indicate that effect size was found in the indices of anxiety and HbA1c by virtue of an ACT intervention [40], and perceived stress was ameliorated through a modified version of MBSR [42]. Regarding the impact that the interventions seemed to have, we found differences by age group: the interventions appeared to have a more significant impact on adolescents aged 15 and older. This would make sense when considering that the ability to be mindful develops with age. Regarding gender, which was balanced in the studies explored, no significant differences were observed with respect to the effect of the interventions studied.

### 4.1. Strengths and Limitations

The main strength to highlight of this review was that it was the first systematic review to our knowledge with the purpose of exploring and summarizing the impact of programs based on mindfulness for young populations with T1D. Additionally, we were open to the incorporation of various programs under the broad banner of mindfulness to achieve a broader vision of the possibilities of these types of programs or intervention packages that incorporate exercises based on mindfulness. MBSR and ACT adaptations were shown to have the largest effect sizes.

Regarding the main methodological strengths of the reviewed study designs, eight of the nine included groups used an interventional design with a control group, and the distribution by sex was quite balanced, with 44.1% of the samples being women. Another strength to highlight has to do with the time of the interventions described in the studies, since they covered an average of 9.89 weeks. Thus, the average fit the standard protocol of the main original interventions of mindfulness, recommended to run 8 weeks. Regarding adherence to the interventions, the included studies reported a retention rate of 77.1%. And regarding the variables analysed, most of the studies [40,41,42,46,47,48] assessed both psychological and biomedical variables.

One of the main limitations detected during this review was related to the average sample size, which consisted of an average of 35 participants in the different studies. Also, we found a relatively small RCT, which requires a more careful study involving quantitative methodology and comparative intervention groups with randomized control groups. In addition, it was possible that by adjusting MBI session times, the final effects were reduced. Another main limitation was that the vast majority of the studies (66.6%) were pilot studies, and only one study included a follow-up measurement [42]. Likewise, another limitation had to do with a follow-up period; thus, we recommend extending the phases of post-periods to possibly capture longer-term effects (i.e., those beyond 6 months). Despite these limitations, we found that 66.6% of the studies included in this review had a high or acceptable level of methodological quality. These limitations might be deliberately overcome in future studies to further enrich this field of knowledge.

### 4.2. Practical Implications and Future Lines of Research

As the results analysed indicated, most of the studies reviewed were carried out in the last 5 years, so it could be assumed that this field of knowledge is beginning to flourish. Hopefully, the coming years will witness more interventional studies that use MBIs, allowing researchers to expand on our systematic review and perhaps provide methodological improvements. In short, regarding the practical implications, we recommend giving increased visibility to the possible potential that these types of interventions can have in terms of improving the psychological and biomedical symptoms experienced by the target population to eventually achieve a unified intervention protocol for integration in the near future, also adhering to the good practice guidelines and recommendations considered effective for the control of T1D in young populations.

We hope future lines of research address this topic and reinforce the results for T1D. Furthermore, we imagine investigations into the impact of programs based on mindfulness for other young populations at risk, like children with obesity, studying the impact of the practices of conscious eating or mindful eating. Likewise, one might combine both young populations, those with a diagnosis of diabetes and obesity, given that the conditions tend to be associated. Despite obesity being commonly linked to T2D, obesity may also be associated with T1D in a bidirectional way [49]. Therefore, mindfulness practices, especially mindful eating—the main objective of which is to establish a healthy and mindful relationship with food, understood as a trend that tries to reconcile emotions with the moment of eating—as some studies in this regard already indicate [50,51,52], could generate promising benefits, both in psychological and biomedical variables. Finally, regarding the variables explored, low levels of stress related to diabetes management (e.g., DD) may be associated with lower HbA1c. In this line, future research might consider the potentially greater impact of diabetes on the mental health of younger people, determining which psychosocial interventions best change behaviour and improve health [53].

## 5. Conclusions

Overall, this systematic review’s purpose was to explore the effectiveness of programs based on mindfulness regarding their impact on psychological and biomedical variables in young populations with T1D. Depending on the type of program, larger effect sizes were seen with MBSR and ACT adaptations. The results are promising and hopefully help inspire new routes of study for the reduction of DD, which is so common in young people with T1D. The main strengths of the MBIs were associated with the periodic practice of mindfulness-based exercises, which had a positive effect on psychosocial (anxiety, perceived stress, DD, self-efficacy, depression, and psychological well-being) and biomedical variables (HbA1C, mean blood glucose, and diastolic BP) in young populations with T1D. Regarding the practical implications of the findings, we recommend increasing visibility to the possible potential of MBIs for improving psychological and biomedical symptoms in young people with T1D, which might help build a unified intervention protocol in the near future that could be integrated into good practice guidelines and recommendations considered effective for the control of T1D in young people. We hope future investigators review the present study and address the main limitations found (e.g., small sample sizes and a small RCT). Additional scrutiny of the studies in this review might bring additional positive results and improvements to a unique and important field of study. Future investigators who continue this line of work might replicate the early results regarding benefits of MBIs for people with T1D. Finally, we recommend researching the impact of programs based on mindfulness for other young populations increasingly at risk (e.g., individuals with T2D or obese individuals). In a high percentage of cases, causes and symptoms are correlated, and researchers might discover the positive impact of MBIs on mindful eating, for example. 

## Figures and Tables

**Figure 1 healthcare-12-01876-f001:**
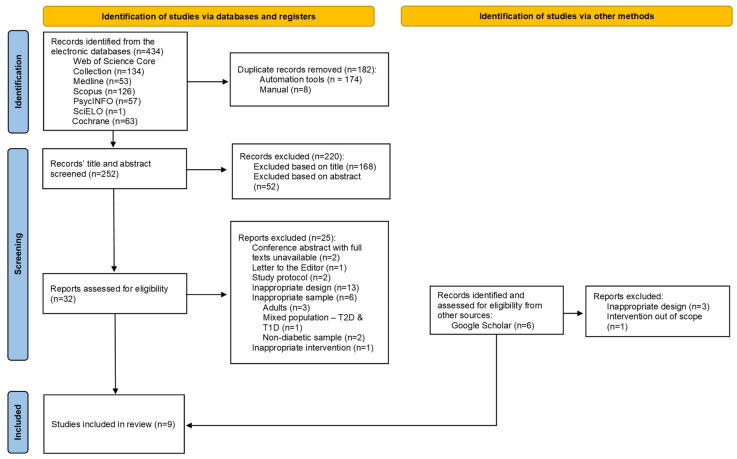
PRISMA flow chart of the selection process. Note. T1D = Type 1 Diabetes; T2D = Type 2 Diabetes.

**Table 1 healthcare-12-01876-t001:** PICOS question for this systematic review.

Participant—Young people with T1D (aged 10–24 years) Intervention—Mindfulness-based interventions (e.g., MBIs, MBSR, MBCT, and MB-EAT) Comparison—Control group versus experimental group; experimental group before–after the intervention Outcome—Psychological outcomes (primary outcome): emotional well-being, anxiety and depression, self-efficacy, diabetes distress, and HRQoL; biomedical outcomes (secondary outcome), e.g., HbA1c and time in range Study design—Primary interventional studies

Note: T1D = type 1 diabetes; MBIs = mindfulness-based interventions; MBSR = Mindfulness-Based Stress Reduction; MBCT = Mindfulness-Based Cognitive Therapy; MB-EAT = Mindfulness-Based Eating Awareness Training; HRQoL = health-related quality of life.

**Table 2 healthcare-12-01876-t002:** Selection criteria.

Inclusion Criteria	Exclusion Criteria
Type 1 diabetes	
Quantitative studies (RCTs, NRCTs, and pre–post without control group)	Type 2 diabetes and other chronic disease population
Studies with a sample comprising individuals within the age range of 10 to 24 years.	Studies with a sample of individuals aged 25 years and older
Original research published (1) in English, Spanish, Italian, and Portuguese in a peer-reviewed journal	Conference papers, proceedings, study protocols, reviews, commentaries, books, book chapters, editorials, position papers, and congress abstracts
Year of publication timespan: 2013–2024	Inappropriate intervention

Note: RCTs = Randomized Controlled Trials; NRCT = Non-Randomized Controlled Trials. (1) Empirical studies in published doctoral theses were also considered in this category.

**Table 3 healthcare-12-01876-t003:** Results of the methodological quality assessment of the studies included.

Study	It. 1	It. 2	It. 3	It. 4	It. 5	It. 6	It. 7	It. 8	It. 9	It. 10	It. 11	It. 12	It. 13	It. 14	Tot. Sc.	Qlt.
[40]	1	1	1	0	0	0	1	1	1	1	1	0	1	0	9	Fair
[41]	0	1	1	0	1	0	0	0	0	1	0	0	0	0	4	Poor
[42]	1	1	1	0	1	1	1	1	1	0	1	0	1	1	11	Good
[43]	0	1	1	0	0	1	0	1	0	1	1	0	1	0	7	Fair
[44]	0	0	0	0	0	0	1	1	0	1	1	0	0	0	4	Poor
[45]	0	0	0	0	1	1	1	1	1	1	1	0	0	0	7	Fair
[46]	0	1	1	0	0	1	1	1	1	1	0	0	0	0	7	Fair
[47]	0	0	0	0	1	0	0	1	0	0	1	0	1	0	4	Poor
[48]	1	1	1	0	0	1	1	1	0	1	0	0	-	-	3	Fair

Note: the results are shown for each item (It.), the total score (Tot. Sc.), and the quality (Qlt.).

**Table 4 healthcare-12-01876-t004:** Basic characteristics of the studies.

Author (year)	Country; Study Design	Measure Points	Sample Size Recruitment Setting	Mean Age ± SD (Female Participants ^a^)	Groups	MBI Protocol	Overall Retention Rate in IG (%) ^a^	Pilot Study ^d^	Main Results ^e^
[40]	Finland; 2-arm RCT	Baseline, post-intervention	IG: 31 CG: 29 Paediatric diabetes outpatient clinic	Total: not reported IG: 13.39 ± 1.12 (71%) CG: 13.48 ± 1.30 (55%)	IG: ACT + TAU CG: TAU (follow-up)	ACT-based group program influenced by previous ACT interventions developed at the University of Jyväskylä. Five 1.5 h sessions in groups of five to seven adolescents, with homework, for 8 weeks (1 session every 15 days).	80.55%	Yes	Intra-group (pre–post): pre–post change between groups, Wald’s test + HbA1c (*p* = 0.001, *d* = 0.59) + Psychological flexibility and acceptance of the diabetes (*p* = 0.04, *d* = 0.29) + Anxiety (*p* = 0.012, *d* = 0.48) Inter-group (post): pre–post change between groups, Wald’s test + HbA1c (*p* = 0.001, *d* = 0.37) + Psychological flexibility and acceptance of the diabetes (*p* = 0.04, *d* = 0.26) + Anxiety (*p* = 0.012, *d* = 0.25)
[41]	New Zealand; feasibility study with randomized IG and CG	Baseline, post-intervention	IG: 11 CG: 8 Paediatric and adolescent diabetes outpatient clinics	Total ^b^: 13.9 ± 1.3 (60%) IG: 14.00 ± 1.2 (64%) CG: 13.6 ± 1.3 (38%)	IG: self-compassion intervention CG: wait-list	Two group sessions of 2.5 h delivered 1 week apart. Brief self-compassion intervention adapted from standardized self-compassion programme for adolescents “Making Friends with Yourself”: group exercises and discussions, art activities, mediations, and individual reflection exploring self-compassion.	70%	Yes (feasibility)	Intra-group (pre–post): not applicable, only descriptive analyses were performed Inter-group (post): not applicable, only descriptive analyses were performed
[42]	UUEE; 3-arm RCT	Baseline, post-intervention, 3-month follow-up	IG: 16 CG1: 16 CG2: 16 Endocrinology outpatient clinics	Total: 18.20 ± 1.43 (50%) IG: 18.00 ± 1.5 (63%) CG1: 18.1 ± 1.3 (31%) CG2: 18.5 ± 1.6 (56%)	IG: MBSR CG1: cognitive–behavioural stress management CG2: support group	A modified version of MBSR developed for use with urban youth consisting of nine weekly sessions. Each session lasted approximately 90 to 120 min.	81.25%	Yes	Intra-group (pre–post) + Stress (*p* = 0.03, *d* = 0.49) Intra-group (pre-follow-up) + Stress (*p* = 0.01, *d* = 0.67) Inter-group (post-CG1; post-CG2): not applicable in the absence of comparative analysis Inter-group (follow-up CG1; follow-up CG2): not applicable in the absence of comparative analysis
[48]	UUEE; feasibility study with IG	Baseline, post-intervention	IG: 6 Endocrinology outpatient clinic	IG ^b^: 18.6 ± 1.2 (90%)	IG: MBSR	A modified version of MBSR developed for use with urban youth consisting of nine weekly sessions. Each session lasted approximately 90 min. Group intervention. Instruction and practice of meditation, mindfulness techniques, and group discussions focused on mindfulness. Dysregulated focus on the past (rumination/depression) and worry about the future (anxiety).	60%	Yes (feasibility)	Intra-group (pre–post) + Diabetes stress (*p* = 0.04) + Mean seven-day blood glucose (*p* = 0.03)
[44]	Iran; 2-arm (R)CT	Baseline, post-intervention	IG: 16 CG: 16 Diabetes association	Total: not reported (30.56%) IG: 11.44 ± 2.59 (25%) CG: 9.72 ± 2.37 (43.75%)	IG: ACT CG: control	A modified version of ACT, with sessions of 90 min every week for ten weeks. The topics for each session included creative hopelessness, value clarification and building a commitment, control as a problem, the alternative to control, cognitive diffusion, self as context, acceptance and commitment, and fear of commitment.	88.89%	No	Intra-group (pre-post): not applicable in the absence of comparative analysis Inter-group (post): multiple analysis of covariance + Stress (*p* < 0.01) + Negative stress (*p* < 0.001) + Positive stress (*p* < 0.001) + Special health self-efficacy (*p* < 0.001)
[45]	Iran; 2-arm (R)CT	Baseline, post-intervention	IC: 17 CG: 17 Diabetes association	Total: not reported (50%) IG: 10.35 ± 2.91 (52.94%) CG: 10.59 ± 3.16 (47.06%)	IG: ACT CG: control	A modified version of ACT, with sessions of 90 min every week for ten weeks. The topics for each session included creative hopelessness, value clarification and building a commitment, control as a problem, the alternative to control, cognitive diffusion, self as context, acceptance and commitment, and fear of commitment.	85%	No	Intra-group (pre-post): not applicable in the absence of comparative analysis Inter-group (post): multivariate covariance analysis + Depression (*p* < 0.001) + Psychological well-being (*p* < 0.001) + Feeling of guilt (*p* < 0.001)
[47]	UUEE; 2-arm (R)CT	Baseline, post-intervention, 3-month follow-up	IG: 33 CG: 32 Summer camps for adolescents with diabetes and paediatric endocrinology outpatient clinics	Total: not reported IG ^b^: 15.64 ± 1.9 (48.5%) CG ^b^: 14.81 ± 1.99 (53.1%)	IG: MBSR CG: wait-list	Six-week online/web-based instructional MBSR program adapted to adolescents. Six main modules through the website MySweetMind.org.	73.86%	Yes	Intra-group (pre–post) + Mindful attention awareness (*p* < 0.001) + Diabetes quality of life (*p* = 0.004) Intra-group (pre-follow-up) + Mindful attention awareness (*p* < 0.001) + Diabetes quality of life (*p* < 0.001) Intra-group (post-follow-up) + Diabetes quality of life (*p* < 0.001) Inter-group (post and follow-up): analysis of variance + Mindful attention awareness (*p* < 0.001, *η* = 0.691)
[46]	India; 2-arm (R)CT	Baseline, post-intervention	IG: 16 CG: 16 Endocrinology outpatient clinic	Total: 23.8 ± 6.6 (53.1%) IG: 24.8 ± 8.6 (not reported) CG: 22.8 ± 3.9 (not reported)	IG: breathing exercises and meditation CG: control	Training program with two components, breathing exercises and mediation, and seven steps. They had to practice at least 20 min at home.	100% ^c^	No	Intra-group (pre–post) + Diabetes distress (*p* = 0.003) + Quality of life, health and functioning (*p* = 0.023) + Mean blood glucose (*p* = 0.008) + HbA1c (*p* = 0.008) + Diastolic BP (*p* = 0.002) Inter-group (post): not applicable in the absence of comparative analysis
[43]	Iran; 2-arm quasi-experimental (R)CT	Baseline, post-intervention	IG: 10 CG: 10 Not reported	Total: 14.6 ± 1.8 (not reported) IG: not reported CG: not reported	IG: yoga-based mindfulness training CG: control	Yoga-based mindfulness training on anxiety and depression of ten weeks, 45 min every week.	100% ^c^	Yes	Intra-group (pre–post): *p* not significant in comparison, and the analysis of covariance does not show the significance of *p* Inter-group (post): multivariate analyses + Anxiety, depression, and stress (*p* = 0.0001)

Note: (^a^) The percentage was calculated in those studies that did not report it. (^b^) The age of the sample analysed was not reported, only of the initial sample. (^c^) The data were not explicitly reported in the study. (^d^) The assessment of whether or not it is a pilot study was made on the basis of the authors’ own statements. (^e^) Only significant results are indicated. The + symbol indicates improvements in favour of the experimental MBI condition.

**Table 5 healthcare-12-01876-t005:** Psychological and biomedical variables and the instruments utilized for measurement in the studies included.

Psychological Variables		Psychological Variables Related to Diabetes	
Health-related quality of life [40,46,47]	Revised Children’s Quality of Life Questionnaire (KINDL-R) Quality of Life Index—Generic Version (QLI-G) Diabetes Quality of Life for Youth (DQOLY)	Adherence to diabetes-related self-care behaviours [41]	Self-Care Inventory-Revised Version (SCI-R)
Anxiety and depression [40,42,43,45]	Revised Beck Depression Inventory (RBDI) Reynolds’ Child Depression Scale (RCDS) Center for Epidemiologic Studies Depression Scale (CES-D) The Depression, Anxiety, and Stress Scale (DASS-21)	Diabetes-related distress [41,46]	Problem Areas in Diabetes survey (PAID) Diabetic Distress Score (DDS)
Stress [41,42,44,47]	Perceived Stress Scale (PSS) Diabetes Stress Questionnaire (DSQ) Perceived Stress Reactivity Scale (PSRS)	Psychological flexibility and acceptance of the diabetes [40]	Diabetes Acceptance and Action Scale for Children and Adolescents (DAAS)
Self-compassion [41]	Self-Compassion Scale, Short form (SCS-SF)	Disordered eating behaviour [41]	Diabetes Eating Problem Survey (DEPS-R)
Self-efficacy [44]	Special Health Self-Efficacy Scale (SHSES)	Diabetes management [42]	Diabetes Management Scale (DMS)
Feelings of guilt [45]	Eysenck Feeling of Guilt Scale (EFGS)		
Psychological well-being [45]	Satisfaction With Life Scale (SWLS)		
**Psychological Variables Related to Mindfulness**		**Biomedical Variables**	
Dispositional mindfulness [40,46,47]	Child and Adolescent Mindfulness Measure (CAMM) Mindful Attention Awareness Scale (MAAS) Mindful Attention Awareness Scale- Adolescent (MAAS-A)	Weight [46]	Body Mass Index (BMI)
		Metabolic control [40,41,42,46,47]	Glycaemic control (HbA1c), glucose blood BGM, and BP (systolic and diastolic)

## Data Availability

Data are contained within the article and Appendix A. Further inquiries may be directed to the corresponding author.

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
