# Peer review of "Effectiveness of Mindfulness-Based Interventions with Respect to Psychological and Biomedical Outcomes in Young People with Type 1 Diabetes: A Systematic Review"

_healthcare, 2024, doi:10.3390/healthcare12181876_

Round 1
Reviewer 1 Report
Comments and Suggestions for Authors
Dear Authors,
Thank you so much for your submission. The topic related to mindfulness is very hot and interesting in this decade and most people are aware of their spiritual needs. However, I have some comments and suggestions on the manuscript:
1. Please include the registration number in PROSPERO in the manuscript;
2. Search strategies, data sources, and selection process:
a. Include a set of search strategies as supplementary information;
b. how many researchers to conducting the search?
c. the search was conducted until November 2023, I am afraid that there are some new studies are missing;
3. Table 1. PICOS: For participants, provide the age range of the "young";
4. The authors mentioned the study is to investigate the MBIs on biomedical parameters and relevant psychological variables. However, the outcome in Table 1. PICOS, there is only one parameter related to the biomedical aspect. In addition, if there are more outcomes related to the psychological aspects, it is suggested to mention the biomedical aspect as a secondary outcome.
5. Data collection process: provide information on how many people conduct the data collection process, and how to ensure the data entries are correct?
6. Quality assessment: cite the reference number of the studies in this section
7. Effectiveness of the intervention: Were the improvements in HbA1C, blood glucose levels, and diastolic blood pressure significant improvement or not?
8. Table 4 with the biomedical variables of BMI, however, there is no information related to BMI mentioned in the results and discussion
It is suggested that to review the whole manuscript to align with the tables with the methodology, results, and discussion.
Comments on the Quality of English LanguageThe quality of the English Language is readable.
Reviewer 2 Report
Comments and Suggestions for Authors
General comments
Thank you to the authors for giving me the opportunity to review their interesting paper on mindfulness interventions in youth with Type 1 diabetes. Below, I have outlined my comments on the paper.
Specific comments
Title
L2. 'on Outcomes' is too vague; please replace it with a more appropriate term or remove it.
L12. Avoid using abbreviations in the abstract.
L26. According to the PRISMA Abstract-Checklist, the discussion should include: “9) A brief summary of the limitations of the evidence included in the review (e.g., study risk of bias, inconsistency, and imprecision)”, and “10) A general interpretation of the results and their important implications”.
L27. Three of your keywords are included in the title. Please provide an alternative keyword if possible, using MeSH terms.
L130. If possible, please provide the PRISMA checklist as supplementary material.
L133. To improve the readability of the text, please separate the sections on 'Data Sources,' 'Selection Process,' and 'Search Strategy.' Additionally, review the PRISMA checklist.
L141. Given the limited number of studies included in the present systematic review and that the search was conducted in November 2023, please consider updating the search.
L147. Define in the Methods section the age range encompassed by the term 'youth,' and use appropriate references to support this definition.
Table 2. Specify the age range considered in the inclusion criteria.
L158. Please provide the initials of the researchers.
L191. Please include the references for the studies in the Quality Assessment section.
L200. The authors have not provided the supplementary material for the review. It is not available in the system.
L224. The authors present the main results in a superficial manner. The principal results should be detailed thoroughly in the Results section. Please specify the number of studies that evaluated each variable and provide a comprehensive summary of the intra-group and inter-group differences for each variable.
L241. Be as precise as possible in your discussion when referring to the 'youth population.' If possible, replace this term with 'children,' 'adolescents,' or another more specific term.
L249. The Discussion section should address the main outcomes in depth, rather than merely repeating the information presented in the Results section. Overall, the discussion is superficial and requires a more thorough comparison of results with previous studies. This is a crucial aspect of your systematic review.
L300. Separate the subsections on practical applications and limitations in the Discussion section.
Round 2
Reviewer 2 Report
Comments and Suggestions for Authors
General comments
Comment 1: Thank you for your thorough response to my comments and suggestions. While I believe that most of my concerns have been appropriately addressed, there remain a few aspects that require further attention before the manuscript can be considered for acceptance.
Specific comments
Response: Thank you for this recommendation. The PRISMA checklist is now upload as a supplemental material.
Comment 2: A significant number of items outlined in the PRISMA checklist are either incomplete or unclear. It is concerning that the authors assert adherence to PRISMA guidelines when 13 of the items have not been reported. This raises serious concerns.
Response: Thank you for this appreciation. We have included this information in the Table 1 and in Table 2. We have also specified the accuracy of this age range in the introduction section.
Comment 3: Thank you to the authors for clarifying this issue. However, in Table 2, it is stated: "Studies with a sample of individuals aged 25 years and older." What about individuals younger than 10 years old? Please, clarify.
Response: The data file with the quality assessment has now been upload as supplementary file, together with the PRISMA checklist, the search strategy and the PROSPERO registration.
Comment 4: The “Instructions for Authors” indicate:
"Supplementary Materials: Describe any supplementary material published online alongside the manuscript (figures, tables, videos, spreadsheets, etc.). Please indicate the name and title of each element as follows: Figure S1: title, Table S1: title, etc."
Please ensure this formatting is applied consistently throughout the manuscript.
Comment 5: Kindly consider incorporating the quality assessment tables into the main body of the text, rather than as supplementary material.
